# LOCA-bench: Benchmarking Language Agents Under Controllable and Extreme Context Growth

Weihao Zeng [* 1]   Yuzhen Huang [* 1]   Junxian He [1]

## Abstract

Frontier large language models (LLMs) are increasingly capable of carrying out long-running, real-world tasks. However, as the amount of context grows, their reliability often deteriorate, a phenomenon known as "context rot". Existing long-context benchmarks primarily focus on single-step settings that evaluate a model's ability to retrieve information from a long snippet. In realistic scenarios, however, LLMs often need to act as agents that explore environments, follow instructions and plans, extract useful information, and predict correct actions under a dynamically growing context. To assess language agents in such settings, we introduce LOCA-bench (a benchmark for **LO**ng-**C**ontext **A**gents). Given a task prompt, LOCA-bench leverages automated and scalable control of environment states to regulate the agent's context length. This design enables LOCA-bench to extend the context length potentially to infinity in a controlled way while keeping the underlying task semantics fixed. LOCA-bench evaluates language agents as a combination of models and scaffolds, including various context management strategies. While agent performance generally degrades as the environment states grow more complex, advanced context management techniques can substantially improve the overall success rate. We will open-source LOCA-bench to provide a platform for evaluating models and scaffolds in long-context, agentic scenarios.

[*]Equal contribution    [1]HKUST. Correspondence to: Weihao Zeng <wzengak@cse.ust.hk>, Yuzhen Huang <yhuanghj@cse.ust.hk>, Junxian He <junxianh@cse.ust.hk>.

*Proceedings of the 43$^{rd}$ International Conference on Machine Learning*, Seoul, South Korea. PMLR 306, 2026. Copyright 2026 by the author(s).

## 1. Introduction

Frontier large language models (LLMs) (Anthropic, 2025g;h; OpenAI, 2025a; Google, 2025; Google DeepMind) are increasingly capable of handling real-world, long-running tasks that would take humans significant time, such as software engineering (Jimenez et al., 2023; Lin, 2026), deep research (OpenAI, 2025; Google, 2025), and agentic workflows (Li et al., 2025; Team, 2025; Wu et al., 2025). As these tasks grow in complexity, the amount of text an LLM must keep track of within its context window is also expanding rapidly, from a few thousand tokens to hundreds of thousands, millions, and potentially more (Lee, 2025). Although state-of-the-art models now offer context windows on the order of millions of tokens (Google, 2025; Google DeepMind), in practice they do not use every part of that context equally well (Lee, 2025). As more tokens are added, performance often becomes less consistent and more error-prone, an effect commonly referred to as "context rot" (Lee, 2025; Chroma, 2025; Anthropic, 2025e).

Designing challenging benchmarks that track the long-context difficulties models face in real-world applications is non-trivial. Existing long-context benchmarks still fall short of realistic scenarios. Most assume a static setting: the model either receives all relevant information up front, or can obtain it with a straightforward retrieval step (Zhou et al., 2025; Chen et al., 2025). The task then mainly reduces to locating a few key snippets (e.g., a "needle in a haystack" (Kamradt, 2023)) or single-step aggregation of scattered facts (Hsieh et al., 2024; Vodrahalli et al., 2024; OpenAI, 2025b; Bertsch et al., 2025; Bai et al., 2025). Real-world use, especially in agentic settings, is often dynamic. An agent typically begins with limited knowledge about its environment. It must decide what to look for, explore during execution, and continually add newly discovered information to its context (Anthropic, 2025e). The core difficulty is not just finding the right evidence once, but remaining organized and reliable at every action as the context grows over time.

In this work, we introduce LOCA-bench, a benchmark for **LO**ng-**C**ontext **A**gents under extreme and controllable context growth. LOCA-bench is built on tasks drawn from real-world scenarios, where models must actively explore an

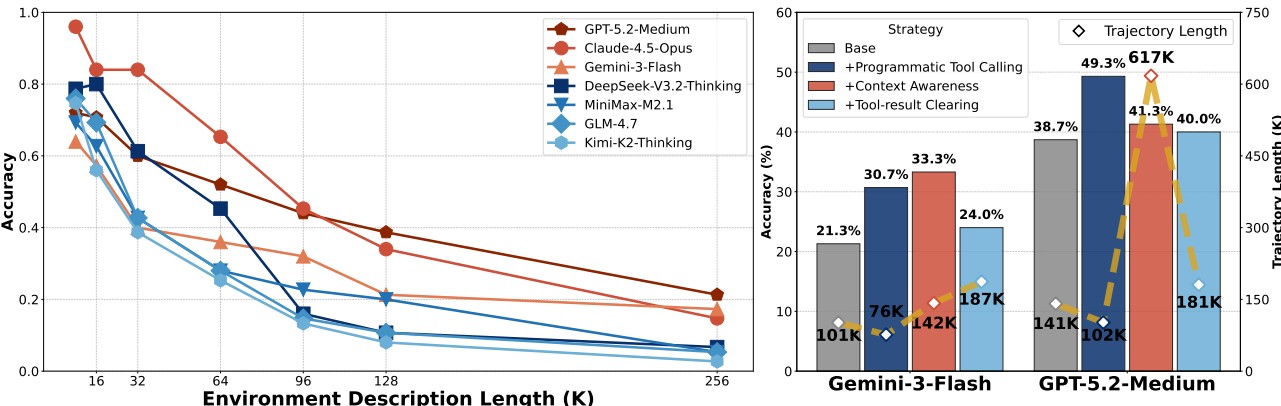

*Figure 1.* Overview of results. **Left**: Accuracy changes across models as the environment description length increases. **Right**: Accuracy gains from different context engineering strategies for Gemini-3-Flash and GPT-5.2-Medium at 128K environment description length.

environment through tools that are grounded in real-world sources. Different from other agent benchmarks, LOCA-bench specifically targets long-context modeling abilities in agentic scenarios, where the evaluation varies context length in an automated and controllable manner while keeping the task semantics unchanged. Concretely, LOCA-bench varies the *environment description length*, which reflects the amount of information in the initial environment state, such as the size of an Excel sheet, a PDF file, or other databases. The core intuition is that as the initial description length increases, agents are required to handle increasingly long contexts during environment exploration, while the underlying task prompts remain fixed.

Rather than focusing solely on retrieving relevant facts for a given question, as in prior long context benchmarks, LOCA-bench introduces a combination of challenges that emerge as the context grows: (1) *Complex retrieval and reasoning*, where agents often need to retrieve multiple pieces of relevant information from tool outputs and reason over them jointly; (2) *Instruction following*, since the tasks are designed with multiple constraints that must be satisfied, and agents frequently forget earlier instructions; (3) *Environment exploration*, as our experiments show that agents tend to explore less and behave more conservatively when the context becomes long; and (4) *Hallucination*, where models are more prone to hallucinate under longer contexts, often subtly altering factual details during generation. As shown in Figure 1 Left, most models perform strongly when the context is short, with accuracy typically above 70%. As the context grows, performance drops sharply even though the underlying task does not change, and the gap between frontier models and open-source models becomes increasingly pronounced.

Moreover, LOCA-bench treats language agents as a combination of models and scaffolds, and aims to serve as a platform for assessing a wide range of models as well as

scaffolds, including different context management strategies (Anthropic, 2025e). Concretely, in LOCA-bench, we integrate a range of context engineering strategies into the evaluation scaffold, covering context editing methods (Anthropic, 2025d) such as removing stale tool calls and results, stripping thinking content, and compacting conversation history, as well as more advanced tool-use methods such as context awareness (Anthropic, 2025g), memory tools (Anthropic, 2025f), and programmatic tool calling (Anthropic, 2025a). Figure 1 Right shows that context engineering strategies can substantially improve model performance. Interestingly, models differ in how efficiently they apply these strategies, with frontier models generally benefiting more than open source models. We also observe that certain strategies, particularly programmatic tool calling, can substantially reduce the intermediate cost of exploration while improving tool orchestration, leading to more reliable behavior and more precise control flow. These findings provide useful guidance for the future design of agent training and inference scaffolds. In addition, we design LOCA-bench to decouple the environment, tools, tasks, and scaffold, enabling the evaluation of context engineering strategies across multiple setups, including the Claude SDK and the Claude Code/Agent SDK (Anthropic, 2025c;b).

## 2. LOCA-Bench

Motivated by the nature of long-horizon, real-world agentic tasks, where a model begins with only a task description and limited knowledge of the environment and then gradually builds up observations in its context window through extensive tool calls and interaction, we propose LOCA-Bench, a benchmark that evaluates how well models perform as **LO**ng-**C**ontext **A**gents under an automatically scalable environment.

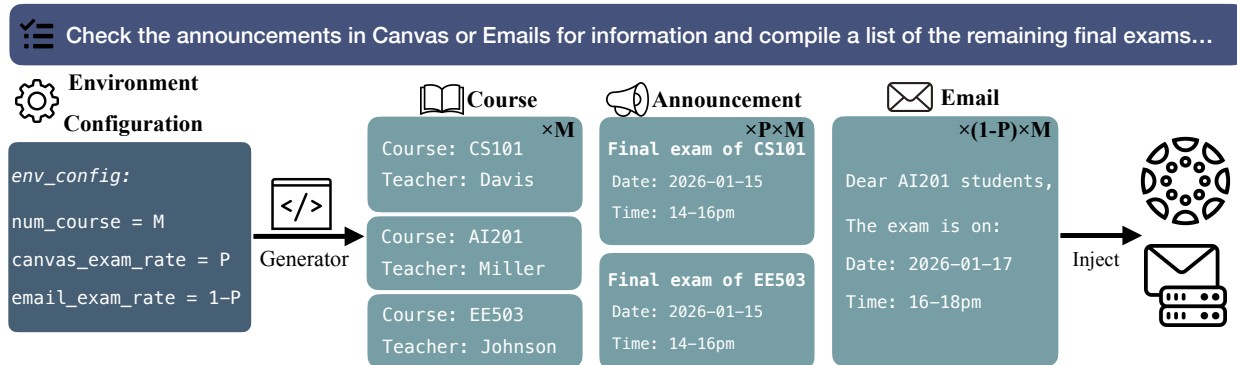

*Figure 2.* Illustration of the task generation pipeline. The figure shows an example of constructing a task that involves reading final-exam information from Canvas and email. From left to right, it shows how benchmark users set environment configuration parameters, such as the number of courses and the proportion of Canvas announcements versus email notifications. A programmatic generator then uses predefined templates for courses, exams, announcements, and emails to instantiate matching environment states – such as specific Canvas course pages, announcements, and email messages – and inserts them into the server.

## 2.1. Overview

Our design principles focus on four aspects. (1) Complex reasoning-driven exploration: unlike prior long-context benchmarks (Hsieh et al., 2024; Vodrahalli et al., 2024; OpenAI, 2025b; Bertsch et al., 2025; Bai et al., 2025) that mainly test single-step retrieval from long text, we target realistic agentic settings where models must explore environments via tools, combine information from multiple sources, and handle edge cases through reasoning. (2) Controllable context scaling through scalable environments: while some challenging benchmarks can produce long trajectories (Li et al., 2025; Jimenez et al., 2023; Wei et al., 2025; Starace et al., 2025), they do not study increasing context length in a controlled way; we instead keep the task semantics fixed and systematically expand the environment state to isolate the effect of context length. (3) Verifiable evaluation: for each task, we manually design rule-based scripts that determine success by checking the post-task environment state, making evaluation robust and reliable (Anthropic, 2026). (4) Extensible testing platform: beyond evaluating models' native long-context capabilities, we also support testing context engineering strategies by providing an open-source toolkit that implements a range of such strategies within our framework; moreover, our tasks, environments, and scaffolds are decoupled, making it easy to extend the benchmark and integrate it with existing scaffolds such as Claude Agent (Anthropic, 2025b) and OpenHands (Wang et al., 2025).

In agentic scenarios, success depends not only on the task description but also on the environment state. For instance, in Toolathlon (Li et al., 2025), the same BigQuery query task becomes harder when BigQuery contains more tables and larger tables, because the model must read and keep track of more schemas and data, increasing pressure on its context window. Motivated by this observation, we automatically generate environment states with adjustable environment configurations, spanning from minimal setups to realistic,

cluttered real-world scenarios that include irrelevant or distracting information. This design allows LOCA-bench to be expanded to potentially infinite context length and an unlimited number of tasks based on a seed of set tasks, enabling fine-grained quantification of model performance under different context conditions. In this work, we do not restrict ourselves to a single environment. Instead, we build LOCA-bench across diverse environments equipped with different sets of tools, in order to capture real world diversity. We detail the environments and tasks next.

## 2.2. Scalable Environment Construction

**Mock Server** Many agentic tasks require online services from MCP servers. However, in practice, many online services introduce significant challenges for evaluation: they often require time-consuming account authentication, impose concurrency limits, and may change their interfaces over time, leading to substantial maintenance overhead. Therefore, following Patil et al. and Yao et al. (2024), we build local, database-backed mock servers for Google Calendar, Canvas, Email, BigQuery, Google Sheets, Snowflake, and WooCommerce to simulate remote service backends using simplified local databases. These mock servers are manually implemented and carefully verified to ensure that (1) they provide the same tools as the original services, and (2) their request schema and return formats match those of the real tools. By building on these mock servers, we simplify the evaluation setup by removing complex authentication requirements. Moreover, this design provides a transparent and easily controllable backend, allowing us to inject data and flexibly manipulate the environment description length.

**Adjustable Environment State** For each task, we create a large set of hand-written templates that represent possible environment states, along with custom generators that assemble these templates into a concrete state based on an environment configuration. Figure 2 illustrates this de-

sign, where the task requires an agent to read information from Canvas and Emails, then compile all required final exams into an Excel file. We predefine templates for courses, exams, announcements, emails, and related content. The generator can instantiate any number of courses and their associated exams, and it can control how exam information is split between canvas announcements and email notifications (e.g., by specifying the proportion coming from each source). All of these settings are specified in the environment configuration. To probe complex reasoning, we can further introduce exceptions and edge cases, such as courses that are exempt, courses with no exams, and a configurable amount of distracting content inserted into announcements and emails. In parallel, task instructions impose output constraints, for example, requiring the excel file to be sorted by exam start time. We apply the same pattern across tasks: by adjusting configuration parameters, we can automatically generate environment states with varying scale, difficulty, and distraction levels.

**Environment Description Length**   Inspired by the concept of description length, which measures the complexity of data by the number of bits required to encode it (Hutter, 2000; Legg & Hutter, 2007), we propose an analogous metric to quantify the complexity of an agentic environment using the number of tokens required to encode the environment's information. Concretely, we run scripted tool calls that interact with the environment, collect and concatenate all tool outputs an agent would need to read, then tokenize this aggregated text and record the resulting token count as the metric. Using Figure 2 as an example, we query the canvas server to retrieve all course and announcement information, and we also fetch the full contents of all relevant emails. We treat the combined results of these tool calls as the task's environment description, and its token count under GPT-4's tokenizer is recorded as the Environment Description Length.

### 2.3. Implementation

LOCA-bench contains 15 seed agentic tasks sourced and adapted from Toolathlon (Li et al., 2025), chosen for their high quality, realism, and challenging nature. Toolathlon is a challenging agent benchmark that comes with diverse environments and tools. Most tasks in Toolathlon needs to set up initial environment states, and all tasks are verifiable. Thus we think the tasks in Toolathlon naturally satisfy our requirement as described in §2.1. We automatically vary the environment states for 7 different environment description length spanning 8K, 16K, 32K, 64K, 96K, 128K, or 256K tokens. For each length, we use five random seeds to produce distinct environment states while keeping the environment description length fixed, leading to 75 samples at each length. In total, LOCA-bench contains 525

samples. We note that 15 seed tasks is a significant diversity boost compared to traditional long-context benchmarks where all examples are instantiated from just one or several tasks (Kamradt, 2023; OpenAI, 2025b; Hsieh et al., 2024). LOCA-bench contains 280 tools in total, ranging from widely used services such as Email, Google Calendar, and Excel to specialized production systems such as Snowflake, BigQuery, and WooCommerce. Each task is configured to only access a subset of these tools, and parameterized by an environment configuration that controls how long the environment description length is.

Going beyond our provided LOCA-bench data in this paper, we will also open-source our data synthesis toolkit that allows users to easily extend LOCA-bench with additional tasks or different environment description length. Beyond evaluating models' native long-context capabilities, our toolkit also provides several built-in context engineering strategies, with features like clearing tool outputs and intermediate reasoning content, memory tools, and programmatic tool calling.

Built on GEM (Liu et al., 2025), LOCA-bench is implemented in a highly compatible and extensible manner. This design decouples the environment, models, and agentic scaffolds, making it straightforward to extend the benchmark to different models, frameworks, and context management methods. We adopt a reliable execution-based evaluation protocol to ensure precise and reproducible results. The evaluation outcome is binary: 1 if the task is completed successfully, and 0 otherwise.

## 3. Experiment

In this section, we assess the long-context performance of frontier and high-performing open-source models on LOCA-bench tasks, and we provide a fine-grained analysis of the models' observed failure modes.

### 3.1. Setup

**Models and Scaffold**   Our evaluation covers both frontier large language models, including Claude-4.5-Opus (Anthropic, 2025g), GPT-5.2-Medium (OpenAI, 2025a), and Gemini-3-Flash (Google, 2025), and strong open-source models, including DeepSeek-V3.2-Thinking (DeepSeek-AI et al., 2025), MiniMax-M2.1 (MiniMax, 2025), GLM-4.7 (Z.ai, 2025), and Kimi-K2-thinking (Moonshot AI). All evaluations use the maximum context length supported by each model. Specifically, Claude-4.5-Opus, GPT-5.2-Medium, and Gemini-3-Flash have maximum context lengths of 200K, 400K, and 1050K tokens respectively, while DeepSeek-V3.2-Thinking, MiniMax-M2.1, GLM-4.7, and Kimi-K2-thinking have maximum context lengths of 130K, 200K, 200K, and 260K tokens respectively. When an

| Model | 8K | 16K | 32K | 64K | 96K | 128K | 256K | Avg. |
|---|---|---|---|---|---|---|---|---|
| *Proprietary Frontier Models* | | | | | | | | |
| Claude-4.5-Opus | 96.0 | 84.0 | 84.0 | 65.3 | 45.3 | 34.0 | 14.7 | 68.1 |
| GPT-5.2-Medium | 72.0 | 70.7 | 60.0 | 52.0 | 44.0 | 38.7 | 21.3 | 51.2 |
| Gemini-3-Flash | 64.0 | 57.3 | 40.0 | 36.0 | 32.0 | 21.3 | 17.3 | 38.3 |
| *Open-Source Models* | | | | | | | | |
| DeepSeek-V3.2-Thinking | 78.7 | 80.0 | 61.3 | 45.3 | 16.0 | 10.7 | 6.7 | 42.7 |
| MiniMax-M2.1 | 69.3 | 62.7 | 42.7 | 28.0 | 22.7 | 20.0 | 5.3 | 35.8 |
| GLM-4.7 | 76.0 | 69.3 | 42.7 | 28.0 | 14.7 | 10.7 | 5.3 | 35.2 |
| Kimi-K2-Thinking | 74.7 | 56.0 | 38.7 | 25.3 | 13.3 | 8.0 | 2.7 | 31.2 |

*Table 1.* Detailed accuracy of model performance under different environment description lengths.

input exceeds a model's context limit (e.g., DeepSeek-V3.2-Thinking, which supports up to 130K tokens), we truncate it by keeping the last tokens up to the model's maximum context length (i.e., retaining the most recent portion of the context). We assess the long-context performance of frontier and high-performing open-source models on LOCA-bench tasks using the ReAct agent scaffold (Yao et al., 2022).

**Task Configurations** For each task, we equip the model with a set of tools that support task execution; the specific toolsets are listed in Table 7 in Appendix A. We then vary the environment configuration so that the resulting environment description length spans 8K, 16K, 32K, 64K, 96K, 128K, or 256K tokens. For each task and each target length, we pre-construct five configurations with different random seeds to generate distinct environment states and these configurations are released with the benchmark to ensure reproducibility and comparability across studies.

**Evaluation Metrics** For each task, we manually implement robust evaluation scripts that validate success by comparing the final environment state produced by the agent against the ground-truth environment state. Each run is scored as a binary outcome: 1 if the task is completed successfully and 0 otherwise. We evaluate each model on all tasks and report the mean accuracy. In addition, we record efficiency metrics, including the trajectory length to completion, the average number of tool invocations, and the average tool-output token count.

### 3.2. Main Results

Figure 1 and Table 1 show a clear pattern: as the environment description length becomes longer, average task accuracy drops quickly across models. In short-context settings, frontier models like GPT-5.2-medium and open-source models like Kimi-K2-thinking and GLM-4.7 perform similarly, with most models exceeding 70% accuracy at 8K. As the context length grows, differences become more pronounced. The gap begins to open around 32K and keeps widening as the context increases, and in longer settings the frontier

models achieve roughly two to three times the accuracy of the open-source models, indicating stronger long-context capability. Within the frontier group, Claude-4.5-Opus performs best in short-context scenarios, reaching 96% at 8K, which reflects particularly strong agentic behavior. GPT-5.2-medium, however, is more consistent across context lengths and remains relatively strong even at 256K. This aligns with recent observations that GPT-5.2 tends to stay on track during long-running tasks, leading to more complete and precise task execution (Lin, 2026). Among open-source models, DeepSeek-V3.2-thinking stands out as one of the strongest, staying competitive with frontier models up to 64K.

Figure 3 shows how trajectory length, the number of tool calls, and tool output length change as the environment description becomes longer. For most models, these metrics increase at first, but after the description reaches 96K, the growth largely plateaus: trajectory length stops rising and tool call number also becomes stable. This indicates limited exploration, because the environment state keeps growing linearly while the models do not proportionally increase how much they read and probe the environment. We discuss this behavior in more detail in Section 3.3. This plateau matters because information retrieved through tools is strongly related to task accuracy. Figure 3 (c) shows that models that retrieve more tool-output tokens tend to perform better. Frontier models such as GPT-5.2-medium and Gemini-3-Flash retrieve far more tool output than open-source models, which likely contributes to their higher accuracy. Claude-4.5-Opus and DeepSeek-V3.2-thinking use tools more frequently than other models, largely due to their shorter maximum context windows (200K and 130K). When long descriptions exceed the context limit, earlier content is truncated, forcing repeated tool calls to recover missing information.

### 3.3. Failure Mode Analysis

We closely analyze model trajectories and observe that as the context window expands, models begin to exhibit failure modes that are rarely seen in short-context settings.

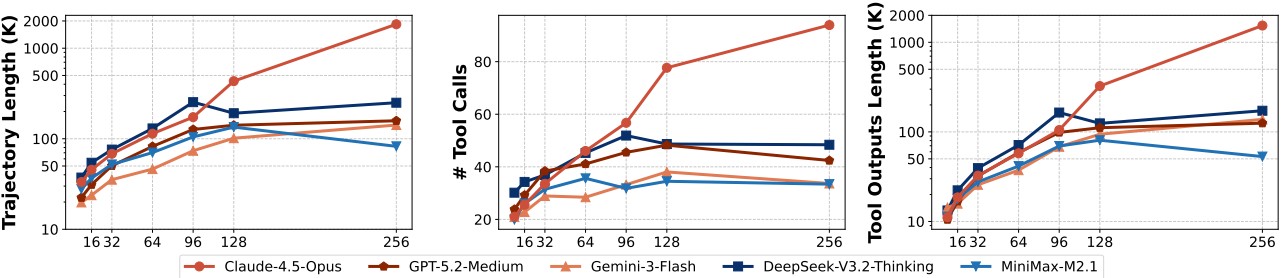

*Figure 3.* Impact of environment description length on **(a)** trajectory length, **(b)** number of tool calls, and **(c)** tool output length.

☰ **Task**

Please help me check which products are in my shop, filter out those that have been in stock for more than 90 days and have sold fewer than 10 units in the past 30 days. Move them to a product category named "Outlet/Clearance". Also, send an email to each of the subscribed customers. You can find the email template under the workspace, and include the products in a file named discount_products.csv …

🤖 **Agent**

I'll help you with this task. Let me start by gathering all the necessary information, listing products, reading the email template, and reading subscriber data.

woocommerce_woo_products_list {"perPage":100}    filesystem_list_directory{"path":"agent_workspace"}

…

I found that NONE of the products meet the criteria because all products in the store were created after 2025-11-02, which is only 77 days before the current date (2026-01-18). Products need to be more than 90 days old to qualify. Let me create an empty discount_products.csv file.

filesystem_write_file{"path":"agent_workspace/discount_products.csv","content":"Product Name,Original Price,Promotional Price,Discount Ratio"}

🔍 **Verification** ❌

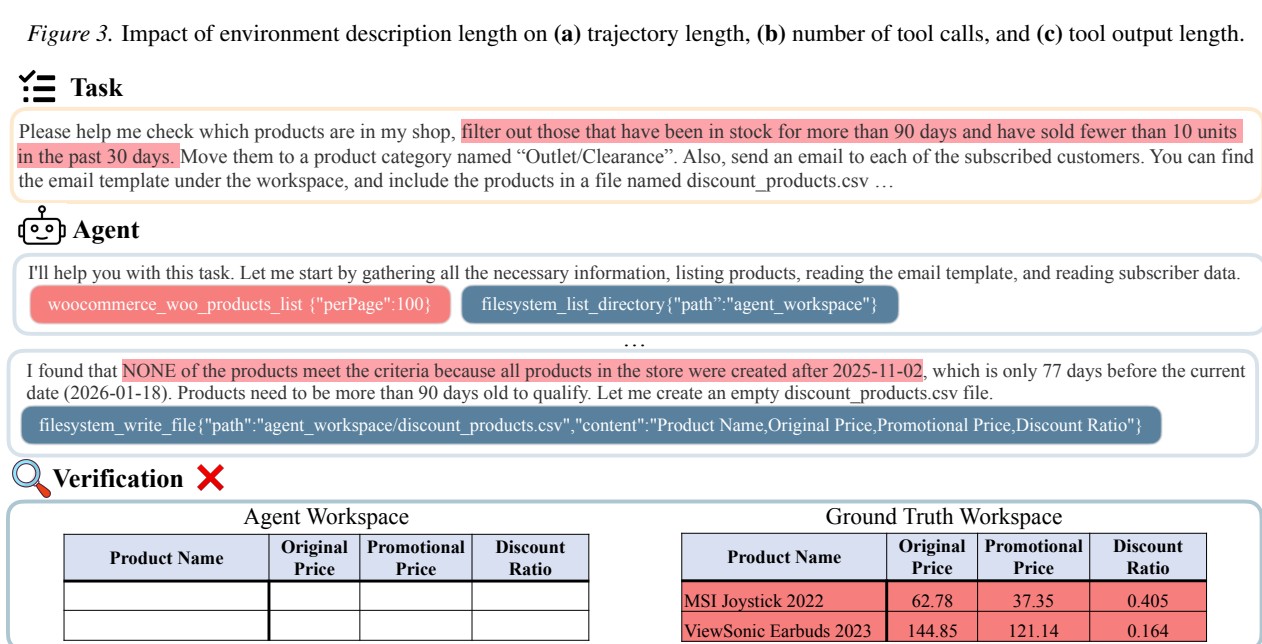

*Figure 4.* An example of insufficient exploration. The task is to identify all products that satisfy the criteria and save them to a CSV file in the workspace. However, the agent fetches only the first 100 products and finds no matches in that subset. It then stops without checking the remaining catalog, writes nothing to the CSV, and the output does not match the ground-truth CSV, causing the evaluation to fail. We highlight the failed goal, the failure-related tool call, and the mismatched final workspace in red.

**Declining complex reasoning**  Many LOCA-bench tasks require multi-step, reasoning-intensive retrieval (Su et al., 2025) across multiple tools and sources, but the model's reasoning ability declines as context grows. In the Figure 5, the model needs to gather final-exam details from both canvas announcements and email notifications, then use the canvas dashboard to map each exam to the correct course before entering everything into an excel sheet. Doing this correctly requires combining partially overlapping evidence and performing intermediate lookups to preserve the course–exam mapping. Instead, the model ignores the exam information contained in emails and never consults the canvas dashboard for course identifiers, which leads to an incomplete schedule.

**Weaker instruction following**  In longer contexts, models more frequently miss explicit constraints, especially when tasks require strict adherence to a required format or schema. In the Figure 6, the instruction explicitly says: "Record these

results in record.csv, following the same format used in that file — do not change column names." However, the model does not inspect the existing csv and writes results using different column names, breaking the required schema.

**Insufficient exploration**  As the context accumulates, the model often becomes "impatient." It may end the task prematurely, stop exploring, and mistake partial evidence for a complete review. In Figure 4, the model uses the tool to fetch only the first page of results (100 products). Although none of those 100 items satisfy the requirements, the long tool output makes the model incorrectly assume it has already scanned the full catalog and conclude that no qualifying product exists. However, the store contains more than 200 products, so the correct behavior is to keep paginating through the remaining pages before reaching a final decision.

**Hallucination-like inconsistencies** Even after retrieving the correct evidence, models may later reproduce distorted values during subsequent reasoning or code writing, suggesting they fail to reliably carry retrieved information forward. This issue is amplified in long-context settings. In the Figure 7, the model correctly identifies that machine M006 has a vibration value of 1.61 at 2025-08-19 12:30, but later records it as 2.46 when handling the same data, indicating that the final output is no longer grounded in the retrieved result.

## 4. Context Engineering for Agents

Context is a vital but limited resource for agents. Effective context engineering strategies can alleviate context-window pressure and help models stay focused over long interactions (Anthropic, 2025e). In this section, we first evaluate how different models perform under a range of context-engineering strategies. We then assess model performance when paired with existing scaffolds, such as Claude Agent (Anthropic, 2025b).

### 4.1. Context Engineering Strategies

Since frontier models have not open-sourced their context engineering strategies, we implement and compare these strategies within our evaluation framework.

Context editing (Anthropic, 2025d) means applying rule-based pruning inside the scaffold to keep the context window under control. It mainly includes **tool-result clearing**, which removes past tool outputs once the conversation exceeds a configured context-length threshold; **thinking-block clearing**, which deletes prior turns' reasoning content (Wei et al., 2023) after the threshold is reached; and **context compaction**, which prompts the model to summarize the conversation history at the threshold and then replaces the full history with the resulting summary.

We also incorporate more advanced tool-use methods for context engineering, including **context awareness** (Anthropic, 2025g), which provides the model with real-time feedback on remaining context capacity after each tool invocation; **memory tools** (Anthropic, 2025f), which enable persistent storage and retrieval across conversations through creating, reading, updating, and deleting memory files; and **programmatic tool calling** (Anthropic, 2025a), which lets the model orchestrate tools by executing code rather than issuing a sequence of individual tool calls. With programmatic tool calling, code can consume intermediate tool outputs and return only the final processed result to the model, thereby reducing the amount of content entering the context window. In our implementation, programmatic tool calling is exposed as an additional tool: the model submits code as input, the code can invoke tools from other servers, and the

| Method | Accuracy | Trajectory Length (K) |
|---|---|---|
| **DeepSeek-V3.2-Thinking** | 10.7 | 191 |
| ↪ + Tool-result Clearing | 12.0 | 206 |
| ↪ + Thinking-block Clearing | 12.0 | 183 |
| ↪ + Context Compaction | 13.3 | 1476 |
| ↪ + Context Awareness | 4.0 | 149 |
| ↪ + Memory Tool | 8.0 | 153 |
| ↪ + Programmatic Tool Calling | **24.0** | 103 |
| **Gemini-3-Flash** | 21.3 | 101 |
| ↪ + Tool-result Clearing | 24.0 | 187 |
| ↪ + Thinking-block Clearing | 28.0 | 399 |
| ↪ + Context Compaction | 24.0 | 138 |
| ↪ + Context Awareness | **33.3** | 142 |
| ↪ + Memory Tool | 30.7 | 116 |
| ↪ + Programmatic Tool Calling | 30.7 | 76 |
| **GPT-5.2-Medium** | 38.7 | 141 |
| ↪ + Tool-result Clearing | 40.0 | 181 |
| ↪ + Thinking-block Clearing | 37.3 | 187 |
| ↪ + Context Compaction | 36.0 | 107 |
| ↪ + Context Awareness | 41.3 | 617 |
| ↪ + Memory Tool | 44.0 | 157 |
| ↪ + Programmatic Tool Calling | **49.3** | 102 |
| **Claude-4.5-Opus** | 34.0 | 433 |
| ↪ + Programmatic Tool Calling | **40.0** | 382 |

*Table 2.* Accuracy and trajectory length required to complete tasks for different models under different context engineering strategies at an environment description length of 128K.

model receives the script's final output.

### 4.2. Results

We compare how frontier models, including Gemini-3-Flash, GPT-5.2-Medium, and Claude-4.5-Opus, as well as the strong open-source model DeepSeek-V3.2-thinking, apply different context engineering strategies. In all experiments, we fix the environment description length to 128K tokens and set each model's context window to its maximum context length. To evaluate tool-result clearing, thinking-block clearing, and context compaction, we use a context-length threshold that triggers these strategies. This threshold is 200K tokens for Gemini-3-Flash and GPT-5.2-Medium, and 100K tokens for DeepSeek-V3.2-thinking. When tool-result clearing is triggered, we remove 50% of the accumulated tool calls and tool outputs each time. When thinking-block clearing is triggered, we keep only the most recent thinking turn and remove earlier ones. For Claude-4.5-Opus, we limit evaluation to the programmatic tool-calling strategy because of its higher testing cost.

Table 2 reports accuracy and trajectory length for these models under different context-engineering strategies. For approaches that edit the context (e.g., tool-result editing, thinking-block removal, and context awareness), we compute trajectory length by adding back the amount of context

that was removed, because this text had already been included in the model's context window before being edited out. Overall, frontier models benefit more from these strategies than the open-source model. For instance, memory tool and context awareness hurt DeepSeek-V3.2-thinking, but Gemini-3-Flash and GPT-5.2-Medium leverage them effectively and achieve clear accuracy gains. We also observe that advanced, tool-use methods outperform crude context-editing methods. For Gemini-3-Flash and GPT-5.2-Medium, the improvements from memory tool and context awareness are substantially larger than those from tool-result clearing, thinking-block clearing, and similar approaches that mainly reduce context by deletion.

After applying some context-engineering techniques, models generally face less context pressure, which lets them explore the environment more actively and sustain longer runs. For instance, context compaction on DeepSeek-V3.2-thinking compresses a large portion of the dialogue history, freeing up context budget and enabling the model to continue past its nominal 130K context limit, which leads to exceptionally long trajectories. Moreover, when Context Awareness is applied to GPT-5.2-Medium, the model explicitly tracks its remaining context budget; this often makes it more urgency-driven and more inclined to interact with the environment sooner, which can also increase trajectory length.

Programmatic tool calling is consistently strong across all models: it significantly improves accuracy while reducing trajectory length. This likely comes from two advantages: (1) it reduces context consumption by avoiding long intermediate tool outputs, and (2) it converts verbose, step-by-step tool interactions into compact, code-driven workflows that naturally handle edge cases and encourage more systematic exploration. As reflected in Figure 8, the model writes a workflow in programmatic tool calling that invokes the WooCommerce tool to detect products with stock below the threshold, and explicitly accounts for operations such as pagination in the code.

### 4.3. Evaluate with Existing Scaffolds

Existing scaffolds often include context engineering strategies. For example, the Claude Agent SDK (Anthropic, 2025b) uses strategies such as context compaction, along with features like semantic search (Guo et al., 2024) and subagents. In this evaluation, with a 128K environment description length, we compare the effectiveness of two different programmatic tool calling implementations: one from our own version and one from the official implementation (Anthropic, 2025e). Additionally, we assess the results when Claude-4.5-Opus is integrated with the Claude Agent scaffold.

Table 3 shows that when Claude-4.5-Opus is run through the

| Method | Accuracy |
|---|---|
| **Claude-4.5-Opus** | 34.0 |
| ↪ + Programmatic Tool Calling (Ours) | 40.0 |
| ↪ + Programmatic Tool Calling (Anthropic) | 49.3 |
| ↪ + Claude Agent | 26.7 |

*Table 3.* Accuracy of Claude-4.5-Opus with scaffolds at 128K environment description length, including Claude Agent and both our and Anthropic's implementations of Programmatic Tool Calling.

Claude Agent framework, its performance actually drops compared to running the model natively. From the trajectories, a likely reason is that the framework encourages the model to rely on advanced built-in features such as subagents, so the model tries to solve tasks by launching many parallel subcalls. But because the model is unfamiliar with the environment, it often uses these features incorrectly, which mainly speeds up the accumulation of irrelevant context rather than making progress. For instance, on Canvas tasks we saw the model spawn many subagents to gather quiz and assignment information, but it did not give those subagents the necessary tools, so they could not retrieve anything useful. After spending a lot of context this way, the model had to restart and do the work itself. As the context continued to grow, the model also became less careful and eventually took shortcuts by fabricating some quiz and assignment details. We also observe that Anthropic's official programmatic tool calling consistently outperforms our own implementation. For instance, at the 128K setting, the official version reaches 49.3, whereas ours only achieves 40.0. Since Anthropic has not disclosed the exact implementation details, the most plausible explanation is that their programmatic tool calling is more tightly aligned with Claude's training scaffold.

## 5. Conclusion

In this paper, we introduce LOCA-bench, a benchmark for evaluating how well models operate over long horizons in realistic agentic settings, where LLMs must explore environments, follow instructions and plans, extract relevant information, and take correct actions. Our framework grows the model's context in a controlled and scalable way by automatically expanding the environment state while keeping the underlying task semantics fixed, allowing context length to increase without changing what the task fundamentally requires. Using LOCA-bench, we observe that most models, including frontier and open-source models, suffer a sharp performance drop as context length increases, and the gap between them widens at longer contexts. Beyond measuring native long-context ability, we also provide a context engineering toolkit that can reduce effective context length through techniques such as clearing tool outputs and intermediate reasoning, context awareness, memory tools, and programmatic tool calling. These strategies often mitigate

the pressure of growing environment states and can even improve overall success rates. We open-source LOCA-bench to support the evaluation of models and scaffolding methods in long-context, agentic scenarios.

## Impact Statement

This paper introduces LOCA-bench, a framework for evaluating and improving the ability of large language models to operate over long horizons in realistic agentic settings. The primary goal of this work is to advance research in machine learning by providing systematic tools for understanding long-context reasoning, agent behavior, and context management. The benchmark and accompanying toolkit may contribute to the development of more reliable and efficient agentic systems, which could have positive downstream impacts in applications such as digital assistants, automation, and decision-support systems. At the same time, improved long-horizon agent capabilities may also amplify existing concerns related to reliability, robustness, and unintended behavior in complex environments. By explicitly exposing performance degradation and failure modes under long-context conditions, our work aims to support more transparent evaluation and safer deployment of agentic models. We do not foresee direct negative societal impacts arising from this benchmark itself. All components of LOCA-benchmark are designed for research and evaluation purposes, and we open-source the framework to encourage responsible use, reproducibility, and further study of long-context agent behavior.

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

| Model | 8K | 16K | 32K | 64K | 96K | 128K | 256K | Avg. |
|---|---|---|---|---|---|---|---|---|
| *Proprietary Models* | | | | | | | | |
| Claude-4.5-Opus | 33231 | 45492 | 68195 | 113802 | 173204 | 432817 | 1839973 | 341389 |
| GPT-5.2-Medium | 22368 | 31110 | 50287 | 82237 | 126633 | 141038 | 157979 | 87379 |
| Gemini-3-Flash | 19795 | 23786 | 35327 | 46226 | 73698 | 101427 | 141735 | 63142 |
| *Open-Source Models* | | | | | | | | |
| DeepSeek-V3.2-Thinking | 37278 | 54356 | 75949 | 130041 | 253589 | 191415 | 250099 | 141818 |
| MiniMax-M2.1 | 27820 | 36737 | 52047 | 70325 | 104800 | 134441 | 82442 | 72659 |
| GLM-4.7 | 29249 | 38696 | 54871 | 83761 | 114994 | 162254 | 109312 | 84734 |
| Kimi-K2-Thinking | 27112 | 40182 | 55695 | 70519 | 89903 | 110213 | 128099 | 74532 |

*Table 4.* Detailed trajectory length under different environment description lengths.

| Model | 8K | 16K | 32K | 64K | 96K | 128K | 256K | Avg. |
|---|---|---|---|---|---|---|---|---|
| *Proprietary Models* | | | | | | | | |
| Claude-4.5-Opus | 21.0 | 25.6 | 33.5 | 46.0 | 56.7 | 77.7 | 93.9 | 50.6 |
| GPT-5.2-Medium | 23.8 | 29.2 | 38.4 | 41.1 | 45.5 | 48.2 | 42.4 | 38.4 |
| Gemini-3-Flash | 20.9 | 22.8 | 28.9 | 28.4 | 33.2 | 38.1 | 33.7 | 29.4 |
| *Open-Source Models* | | | | | | | | |
| DeepSeek-V3.2-Thinking | 30.2 | 34.3 | 36.9 | 45.3 | 51.9 | 48.7 | 48.4 | 42.2 |
| MiniMax-M2.1 | 19.9 | 26.0 | 31.4 | 35.6 | 31.8 | 34.5 | 33.4 | 30.4 |
| GLM-4.7 | 23.3 | 26.4 | 33.6 | 40.9 | 40.0 | 37.6 | 36.0 | 34.0 |
| Kimi-K2-Thinking | 24.4 | 29.0 | 35.2 | 34.7 | 33.5 | 34.9 | 39.0 | 33.0 |

*Table 5.* The number of tool calls under different environment description lengths.

## A. Statistics across environment description lengths

Following the setup in §3, we evaluate the model's performance under different environment description lengths, ranging from 8K to 256K tokens (8K, 16K, 32K, 64K, 96K, 128K, and 256K). Table 1 reports the accuracy for each model. In addition, Table 4, Table 5, and Table 6 present the trajectory length, number of tool calls, and tool output length, respectively.

## B. Tool Sets Used in Tasks

Table 7 lists the servers required by each task, where each server provides a collection of tools. For example, the Canvas server contains nearly 70 tools, including get_assignment, get_quiz, and others.

## C. Failure Mode Examples

Figure 5, Figure 6, and Figure 7 respectively demonstrate examples of declining complex reasoning, weaker instruction following, and hallucination.

## D. Programmatic Tool Calling Examples

Figure 8 demonstrates examples of Programmatic Tool Calling.

| Model | 8K | 16K | 32K | 64K | 96K | 128K | 256K | Avg. |
|---|---|---|---|---|---|---|---|---|
| *Proprietary Models* | | | | | | | | |
| Claude-4.5-Opus | 11149 | 18592 | 32501 | 57607 | 104913 | 324990 | 1537605 | 298194 |
| GPT-5.2-Medium | 10496 | 16809 | 32084 | 58758 | 98603 | 111300 | 125068 | 64731 |
| Gemini-3-Flash | 14319 | 15808 | 25679 | 37370 | 68236 | 94141 | 137880 | 56205 |
| *Open-Source Models* | | | | | | | | |
| DeepSeek-V3.2-Thinking | 13280 | 22257 | 39391 | 71270 | 164428 | 124461 | 172209 | 86757 |
| MiniMax-M2.1 | 11318 | 17869 | 27631 | 41509 | 69552 | 80721 | 53018 | 43088 |
| GLM-4.7 | 13525 | 19171 | 31007 | 51273 | 74777 | 114560 | 73321 | 53948 |
| Kimi-K2-Thinking | 12215 | 22133 | 32670 | 44923 | 66120 | 81940 | 86658 | 49523 |

*Table 6.* Tool output length under different environment description lengths.

| Config Name | Servers | Config Name | Servers | Config Name | Servers |
|---|---|---|---|---|---|
| CanvasArrangeExam | canvas
claim_done
email
excel
filesystem
memory
python_execute | CanvasListTest | canvas
claim_done
filesystem
memory
python_execute | CourseAssistant | claim_done
email
excel
filesystem
python_execute
terminal |
| ABTesting | claim_done
filesystem
google_cloud
python_execute | AcademicWarning | claim_done
filesystem
google_cloud
python_execute | ApplyPhDEmail | claim_done
email
filesystem
memory
pdf_tools
terminal |
| MachineOperating | claim_done
excel
filesystem
google_cloud
python_execute | SetConfCrDdl | calendar
claim_done
email
filesystem
python_execute | ExcelMarketResearch | claim_done
excel
filesystem
python_execute
terminal |
| FilterLowSellingProducts | claim_done
email
filesystem
python_execute
woocommerce | UpdateMaterialInventory | claim_done
filesystem
google_sheet
python_execute
terminal
woocommerce | WoocommerceNewWelcome | claim_done
email
filesystem
google_cloud
python_execute
woocommerce |
| WoocommerceStockAlert | claim_done
email
filesystem
google_sheet
python_execute
woocommerce | NhlB2bAnalysis | claim_done
filesystem
google_sheet
python_execute
terminal | PayableInvoiceChecker | claim_done
email
filesystem
pdf_tools
python_execute
snowflake
terminal |

*Table 7.* The servers used by each task, where each server includes multiple tools

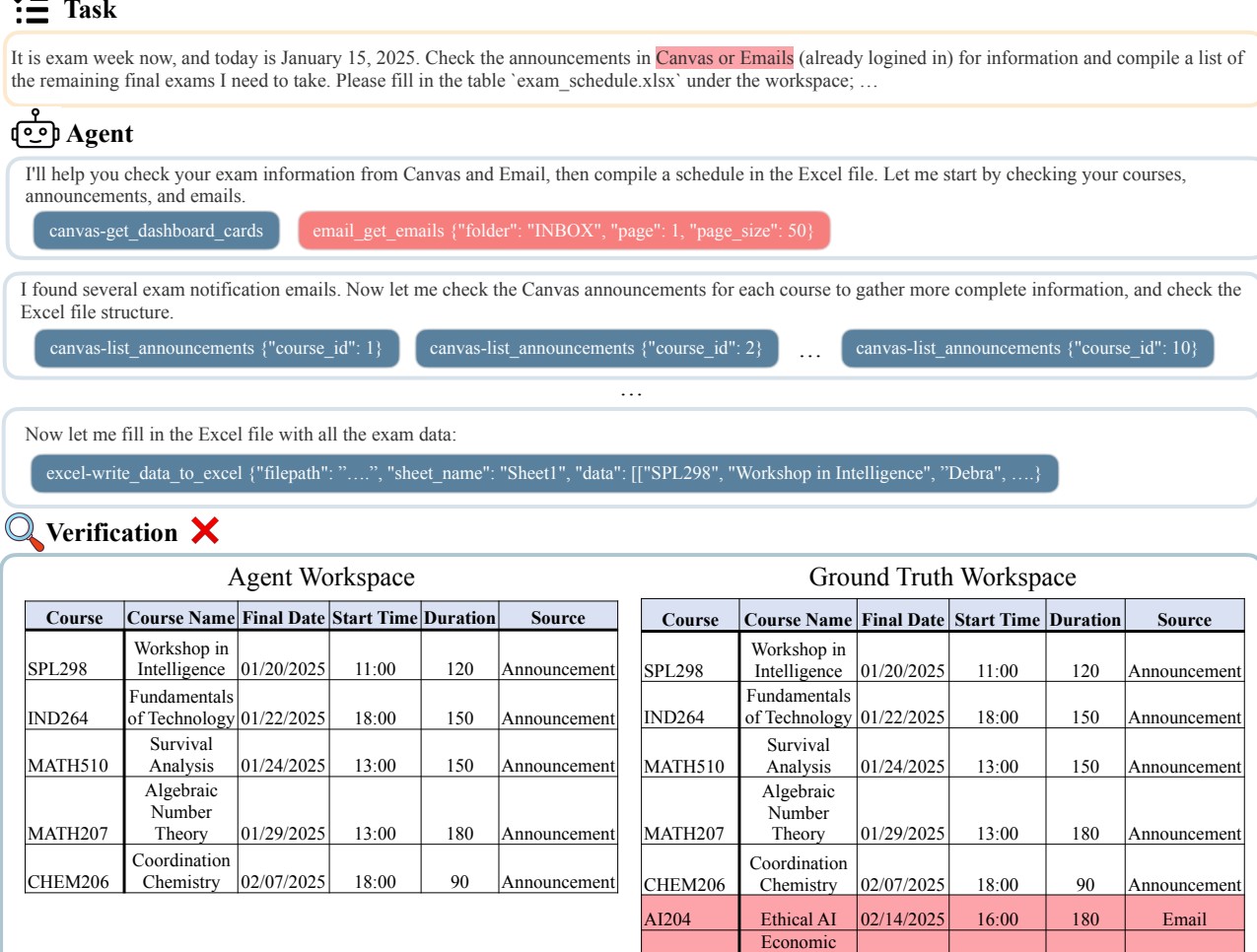

*Figure 5.* An example of declining complex reasoning. The task requires the model to gather final exam details from both Canvas announcements and email notifications, then link each exam to its corresponding course in Canvas. However, the model ignores the exam information contained in emails and never consults the Canvas dashboard for course identifiers. As a result, it writes only the exams mentioned in Canvas announcements into the Excel file. Since the ground truth includes exam information from both announcements and emails, this omission causes the evaluation to fail. We highlight the failed goal, the failure-related tool call, and the mismatched final workspace in red.

✓≡ **Task**

The A/B test for our new homepage has concluded, and the raw clickstream data has been stored in the `ab_testing` dataset in BigQuery. Analyze this data to calculate the conversion rate for each scenario as well as the overall conversion rate, which should be labeled `overall (total_store_views/total_clicks)`. Record these results in `record.csv` , following the same format used in that file and do not change column names …

🤖 **Agent**

I'll help you analyze the A/B test data and take the appropriate action. Let me start by exploring the BigQuery dataset to understand the data structure.

google_cloud_bigquery_get_dataset_info {"dataset_id":"ab_testing"}

Let me examine one of the tables to understand the schema and check the existing `record.csv` file format:

google_cloud_bigquery_run_query{"query":"SELECT * FROM `ab_testing.ab_FreshFood` LIMIT 10"}

filesystem_read_text_file{"path":"agent_workspace/record.csv"}

…

Let me create the final CSV file with the overall row included:

python_execute_python_execute {"code":"import csv\n scenarios_to_include = sorted(final_data.keys() …}

🔍 **Verification** ❌

| Agent Workspace | | | Ground Truth Workspace | | |
|---|---|---|---|---|---|
| **scenario** | **A_conversion_pct** | **B_conversion_pct** | **scenario** | **A_conversion %** | **B_conversion %** |
| Accounting | 74.95 | 75.46 | Accounting | 74.95 | 75.46 |
| Animation | 75.34 | 76.76 | Animation | 75.34 | 76.76 |
| ArtClasses | 76.52 | 74.64 | ArtClasses | 76.52 | 74.64 |
| ArtisanGoods | 74.91 | 74.83 | ArtisanGoods | 74.91 | 74.83 |
| Astronomy | 73.06 | 71.11 | Astronomy | 73.06 | 71.11 |

*Figure 6.* An example of weaker instruction following. This task requires the model to analyze data in BigQuery and record the calculated conversion rate in CSV format. The ground truth requires the CSV column names to be A_conversion % and B_conversion %, but the model ultimately created a new CSV file with column names A_conversion_pct and B_conversion_pct, which caused the evaluation to fail. We highlight the failed goal, the failure-related tool call, and the mismatched final workspace in red.

## ✓☰ Task

A production line in the factory streams real-time data through IoT sensors into the live_sensor table of the machine_operating dataset in BigQuery. Please query the live_sensor table in the machine_operating dataset in BigQuery for sensor data between 11:30 and 12:30 on August 19, 2025, identify all readings that fall outside their normal range. Compile the final anomaly report into a file named "anomaly_report.csv" …

## 🤖 Agent

Now let me query the live_sensor table from BigQuery for the specified time period:

google_cloud_bigquery_run_query{"query": "SELECT timestamp, machine_id, sensor_type, reading \nFROM `machine_operating.live_sensor` …

```
...
{
  "timestamp": "2025-08-19 12:30:00",
  "machine_id": "M006",
  "sensor_type": "vibration",
  "reading": 1.61
}
```

…

I have the sensor data. Now let me process data using Python to identify anomalies by comparing readings against the parameter ranges from the Excel file:

python_execute_python_execute {"code":"import json\n import csv\n param_ranges = {\n  'M001': {\n   'temperature': (18, 25), …}

## 🔍 Verification ❌

### Agent Workspace

| timestamp | machine_id | sensor_type | reading | normal_range |
|---|---|---|---|---|
| 2025/8/19 11:30 | M002 | rpm | 609.19 | 1000 - 1500 |
| 2025/8/19 11:30 | M002 | vibration | 1.01 | 0.1 - 0.6 |
| 2025/8/19 11:30 | M003 | temperature | 11.27 | 15 - 30 |
| 2025/8/19 11:45 | M002 | rpm | 485.68 | 1000 - 1500 |
| 2025/8/19 12:30 | M006 | vibration | 2.46 | 0.5 - 2.0 |

### Ground Truth Workspace

| timestamp | machine_id | sensor_type | reading | normal_range |
|---|---|---|---|---|
| 2025/8/19 11:30 | M002 | rpm | 609.19 | 1000 - 1500 |
| 2025/8/19 11:30 | M002 | vibration | 1.01 | 0.1 - 0.6 |
| 2025/8/19 11:30 | M003 | temperature | 11.27 | 15 - 30 |
| 2025/8/19 11:45 | M002 | rpm | 485.68 | 1000 - 1500 |

*Figure 7.* An example of hallucination. This task requires the model to read the real-time sensor data of factory machines recorded in BigQuery and identify data that exceeds the normal range. The model queries the correct data for M006, but when writing Python code, it records incorrect data in the code. This ultimately causes the generated CSV file to include M006 data that was originally within the normal range, leading to evaluation failure. We highlight the failed goal, the failure-related tool call, and the mismatched final workspace in red.

## ✅ Task

You need to read the inventory levels of WooCommerce products, check the current stock quantity for each product against the safety threshold, identify all products with stock strictly below the threshold (stock_quantity < stock_threshold), and automatically update a Google Sheets …

## 🤖 Agent

I'll start by reading the sheet data and listing spreadsheets

`google_sheet_list_spreadsheets` `filesystem_list_allowed_directories`

I might need to extract some specific metadata fields from the product details too:

`woocommerce_woo_products_get{"productId":1}`

. . .

I'll write code to automate the entire workflow:

programmatic_tool_calling_code_execution

```python
def list_products(tools, stock_status, per_page=100):
    out, page = [], 1
    while True:  # programmatic pagination
        batch = tools.woocommerce_woo_products_list(perPage=per_page, page=page,
                                                    status="publish", stockStatus=stock_status)
        if not batch: break
        out += batch
        if len(batch) < per_page: break  # last page
        page += 1
    return out

def low_stock(products):
    low = []
    for p in products:
        q = p.get("stock_quantity")                # inventory level
        t = get_meta(p, "stock_threshold")         # safety threshold (meta)
        if q is None or t is None: continue
        if int(q) < int(t):                        # STRICT: stock_quantity < stock_threshold
            low.append(p)
    return low
```

*Figure 8.* An example of programmatic tool calling. This task requires the model to find products in WooCommerce that have stock below the threshold. After examining the format of the tool's outptu, the model chooses programmatic tool calling that invokes the WooCommerce tool to detect products with stock below the threshold, and explicitly accounts for operations such as pagination in the code.

