# OpenReview forum: "LOCA-bench: Benchmarking Language Agents Under Controllable and Extreme Context Growth"
_ICML.cc/2026/Conference — ICML 2026 regular_

### Official Review · Reviewer_bNGh · 2026-03-13

**Soundness:** 3
**Presentation:** 3
**Significance:** 3
**Originality:** 2
**Overall Recommendation:** 4
**Confidence:** 1

**Summary:**

The paper introduces LOCA-bench, a benchmark to evaluate long-context language agents in realistic and dynamic environments.

LOCA-bench uses automated control of environment states to scale context length in a controlled way while keeping the underlying task semantics fixed.

Results show that agent performance declines as context grows more complex, but advanced context management strategies improve overall success rates.

**Compliance With Llm Reviewing Policy:**

Affirmed.

**Final Justification:**

My concerns have been adequately addressed. I keep my score as a positive assessment.

**Key Questions For Authors:**

Q. how many people involved to make hand-written templates? and how the quality can be verified?

Q. Can you provide how diverse the dataset is and similar to the real-world environments?

**Limitations:**

I don't see any specific limitations.

**Strengths And Weaknesses:**

Strengths:

The benchmark seems valuable as a practical framework for evaluations in a context-engineering toolkit (e.g., memory tools, context awareness, and programmatic tool use)

The paper is offering some insights into scaling behavior rather than just reporting raw benchmark scores.

Weaknesses:

I don't see any critical weaknesses. However, since this is a benchmark paper and out of my research field, I might miss that the paper can overclaim especially how diverse and representative the environments are.

---

> ### Author Rebuttal · Authors · 2026-03-30
>
> **Q1**: how many people involved to make hand-written templates? and how the quality can be verified?
>
> **A1**: In total, two people spent three months building all 15 tasks and the corresponding hand-written templates.
> We conducted strict manual checks during task design to ensure that each task remained solvable under different environment description lengths. To further verify robustness, during the rebuttal we reran the 32K-setting experiments for DeepSeek-V3.2-Thinking and MiniMax-M2.1 by rebuilding the environment from scratch on different Linux systems. For each model, we added two new runs and compared them with the original run reported in the paper. As shown below, the variance in success rate across runs is small:
>
> | Model                   | Run 1 | Run 2 | Run 3 | Mean  | Std (↓) |
> |-------------------------|------:|------:|------:|------:|--------:|
> | DeepSeek-V3.2-Thinking |  61.3 |  60.0 | 60.51 |  60.6 |    0.53 |
> | MiniMax-M2.1           |  42.7 |  41.3 |  42.7 | 42.23 |    0.66 |
>
> **Q2**: Can you provide how diverse the dataset is and similar to the real-world environments?
>
> **A2**:
> Compared with existing long-context benchmarks, our benchmark places much greater emphasis on real-world agent tasks. For example, OOLONG mainly evaluates classification and relatively simple question answering, while benchmarks such as LongBench v2 focus on document QA, dialogue understanding, code understanding, and structured-data understanding. Although these tasks require long-context reasoning, they are still primarily natural-language understanding tasks. By contrast, our benchmark is designed around tasks in which agents must interact with realistic tools and systems to complete multi-step workflows, making it more representative of real-world agent use cases.
>
> Our task set covers a broad range of scenarios and server combinations, capturing a much wider variety of real-world interaction patterns. It includes tasks that require retrieving and organizing information from multiple sources, such as canvas_arrange_exam, as well as tasks that combine quantitative analysis with conditional decision-making. For example, in ab-testing, the agent must compute conversion rates for each scenario from BigQuery data and use those results to determine the winner. We also include tasks that combine cross-system state synchronization with constrained optimization. In update_material_inventory, the agent monitors new WooCommerce orders, retrieves the Bill of Materials (BOM) from Google Sheets, calculates raw material consumption, updates inventory levels, and then determines the maximum producible quantity for each finished product by applying a minimum-over-materials constraint. In addition, the benchmark contains tasks that require processing unstructured documents and generating new content. In apply-phd-email, for example, the agent must read workspace materials, extract and organize relevant information, and then draft an email. Finally, some tasks require temporal reasoning across multiple sources, such as the cross-source anomaly detection task in machin-operating.

---

> > ### Author Rebuttal · Reviewer_bNGh · 2026-04-06
> >
> > My concerns have been adequately addressed. I keep my score as a positive assessment.

---

### Official Review · Reviewer_nxpc · 2026-03-13

**Soundness:** 2
**Presentation:** 3
**Significance:** 3
**Originality:** 2
**Overall Recommendation:** 3
**Confidence:** 2

**Summary:**

This paper introduces LOCA-bench, a benchmark designed to evaluate large language model agents under extreme and controllable long-context settings. Unlike prior long-context benchmarks that focus primarily on static tasks, LOCA-bench evaluates agents performing environment exploration, tool use, and multi-step reasoning while the context grows. Experiments show that agent performance degrades significantly with increasing context length, while some context management strategies can substantially improve success rates.

**Compliance With Llm Reviewing Policy:**

Affirmed.

**Final Justification:**

My concern regarding W1 still remains, and I believe this is an important distinction to make in the first place. However, it is not considered in the experiments nor well discussed, and therefore I am hesitant to raise my score. That said, I would be fine with the paper being accepted and leave the final decision to AC. Thank you.

**Key Questions For Authors:**

- *When an input exceeds a model’s context limit (e.g., DeepSeek-V3.2- Thinking, which supports up to 130K tokens), we truncate it by keeping the last tokens up to the model’s maximum context length (i.e., retaining the most recent portion of the context).*  -> for how much portion of the experiments for each model is this truncation necessary? I wonder how much this influence the final performance given the task designs.
- *we build local, database-backed mock servers for Google Calendar, Canvas, Email, BigQuery, Google Sheets, Snowflake, and WooCommerce to simulate remote service backends using simplified local databases.* -> is this newly implemented for this paper, or is it a part of Toolathlon?
- The example labeled as “insufficient exploration” (Figure 4) appears closer to a tool-use or pagination reasoning failure rather than a genuine exploration failure. In the example, the agent calls woocommerce_woo_products_list {"perPage":100} and stops after finding no matches in the first 100 products. However, this behavior may stem from a misunderstanding of the tool’s pagination semantics (e.g., not realizing that additional pages must be queried) rather than prematurely terminating exploration. A more canonical exploration failure would involve sequentially inspecting multiple items (e.g., iterating over emails) and stopping early despite remaining unexplored items. Clarifying this distinction would strengthen the interpretation of failure modes.
- Are all the experiments done with zero-shot?

**Limitations:**

yes

**Strengths And Weaknesses:**

Strengths
- The paper experimentally show some interesting points like "*Environment exploration, as our experiments show that agents tend to explore less and behave more conservatively when the context becomes long*", kind of like a systematic way of showing many common agent behaviors.
- The proposed environment description length is an interesting metric.

Weaknesses
- The benchmark measures context complexity primarily through environment description length (i.e., the total tokens in tool outputs describing the environment). However, the authors did not specifically consider some more aspects that may influence the agent's performance. For example, whether it is a chain of many many interaction or a few very long interactions, may actually influence the performance, especially for somes types of context engineering strategies you tested like clearing the output of the tool calling.
-  Limited failure analysis and the limited task difficulty itself makes this more important. The paper qualitatively describes several failure modes (e.g., insufficient exploration, reasoning errors, instruction-following mistakes, and hallucination-like inconsistencies), but does not provide a systematic or quantitative breakdown of these categories. As a result, it remains unclear which factors primarily drive performance degradation under long contexts, and how failure patterns differ across models.  For example, with good few-shot demos (which I think the experiments don't have, seems to be zero shot), the coding aided agent should not have any issue with exploration as the proposed tasks look like that can be dealt with exhaustive search, which gives very clear clue about what to check just like Alfworld kind of environment, but the overall performance is still super bad. For how much or the portion does it fail due to failure to locate the specific information in the gathered information (which again, goes to long context understanding problem) and for how much it is due to some wrong action applied to formalize the final answer or something else?

---

> ### Author Rebuttal · Authors · 2026-03-30
>
> **Q1**: Whether it is a chain of many many interaction or a few very long interactions, may actually influence the performance, especially for somes types of context engineering strategies you tested like clearing the output of the tool calling？
>
> **A1**:  Thank you for the suggestion. Our trajectory analysis suggests that context pressure usually comes from both long interaction sequences and large tool outputs, rather than only one. Thus, while the distinction between “many interactions” and “a few very long interactions” is meaningful, most benchmark tasks do not fall cleanly into either category.
>
> Only two tasks lean more clearly to one side: PayableInvoiceChecker is mainly driven by many short tool calls, while ExcelMarketResearch is driven more by very large tool outputs. However, both are still long-context tasks under the environment description length view, and we do not observe context-engineering strategies that help only one type but not the other.
>
> **Q2**: Limited failure analysis and the limited task difficulty itself makes this more important. The paper qualitatively describes several failure modes ..... but does not provide a systematic or quantitative breakdown of these categories.
>
> **A2:**
>
> In the paper, failure modes were identified mainly through manual inspection. To strengthen this, we add a quantitative analysis in the rebuttal. For each failure pattern, we report the percentage of tasks in which it appears and the percentage of tasks in which it is the primary cause of failure, for both GPT-5.2-medium and MiniMax-m2.1. The results show that these patterns are common across tasks, especially declining complex reasoning, weaker instruction following, and insufficient exploration. They also vary by model: for example, hallucination is rare in GPT-5.2-medium but much more common in MiniMax-m2.1.
>
> **GPT-5.2-medium**
>
> | Failure pattern              | % Tasks | % Primary |
> | ---------------------------- | ------: | --------: |
> | Declining complex reasoning  |   46.7% |     33.3% |
> | Weaker instruction following |   60.0% |      6.7% |
> | Insufficient exploration     |   66.7% |     40.0% |
> | Hallucination                |    6.7% |      0.0% |
>
>
> **MiniMax-m2.1**
>
> | Failure pattern              | % Tasks | % Primary |
> | ---------------------------- | ------: | --------: |
> | Declining complex reasoning  |   80.0% |     33.3% |
> | Weaker instruction following |   40.0% |     13.3% |
> | Insufficient exploration     |   66.7% |     33.3% |
> | Hallucination                |   46.7% |      6.7% |
>
>
> **Q3**: for how much portion of the experiments for each model is this truncation necessary? I wonder how much this influence the final performance given the task designs.
>
> **A3**:
>
> We applied truncation only when the context approached the model’s maximum length. To quantify its effect under the 128K setting, we estimated accuracy without truncation, and also measured how often truncation was triggered. We report results for DeepSeek-V3.2-Thinking (130K context window), MiniMax-M2.1 (200K), and GPT-5.2-Medium (400K). Truncation helped models with smaller context windows, improving accuracy from 5.3 to 10.7 for DeepSeek-V3.2-Thinking, but had no effect on GPT-5.2-Medium (38.7 with or without truncation).
>
> | Model                  | Accuracy | Accuracy w/o truncation | Truncate rate |
> | ---------------------- | -------: | ----------------------: | ------------: |
> | DeepSeek-V3.2-Thinking |     10.7 |                     5.3 |         53.3% |
> | MiniMax-M2.1           |     20.0 |                    16.0 |         28.0% |
> | GPT-5.2-Medium         |     38.7 |                    38.7 |          9.3% |
>
> **Q4**: Is this mock server newly implemented for this paper, or is it a part of Toolathlon?
>
> **A4**: This is a new implementation introduced in our paper (Lines 134–149).  As noted in our response to Reviewer CcVw’s Q3, we also verified that the mock servers remained consistent with the real services.
>
>
> **Q5**: A more canonical exploration failure would involve sequentially inspecting multiple items (e.g., iterating over emails) and stopping early despite remaining unexplored items.
>
> **A5**: We plan to include the following trajectory example to clarify this point. The agent searched for emails related to “Camera-Ready,” and the API returned total=160 and total_pages=8. It explicitly noted that it “need[s] to make 8 calls,” showing that it understood the pagination structure. However, it then chose to stop early, saying that “160 items feels like a lot” and deciding to “focus just on the first page.” Thus, the failure was not due to misunderstanding the tool, but to prematurely stopping exploration despite knowing that more content remained. As a result, it created reminders for only 35 of the 64 target conferences, missing 45% of them.
>
> **Q6**: Are all the experiments done with zero-shot?
> **A6**: Yes. We ran all experiments in a zero-shot setting, which is a general setting for agent.

---

> > ### Author Rebuttal · Reviewer_nxpc · 2026-04-04
> >
> > Thank you for the rebuttal and the additional clarifications. However, my primary concern regarding W1 still remains. Specifically, the distinction between long observations and long interaction sequences is not clearly addressed. These two aspects correspond to fundamentally different sources of difficulty, therefore can lead to different behaviors and require different capabilities from the agent. Without explicitly analyzing these two aspects, it is difficult to interpret what exactly drives the observed performance degradation. I believe this distinction should be discussed or evaluated more carefully. That said, I am open to further discussion if other reviewers or the AC have a different perspective. For now, I will keep my current score.

---

> > > ### Author Response · Authors · 2026-04-06
> > >
> > > **Q1**：my primary concern regarding W1 still remains. Specifically, the distinction between long observations and long interaction sequences is not clearly addressed.
> > >
> > > **A1:**
> > > We thank the reviewer for the follow-up. We agree that long interaction sequences and long per-call observations are distinct sources of difficulty. Consistent with A1, however, most LOCA-bench tasks do not fall cleanly into only one category: in our analysis, most tasks receive substantial contributions from both interaction count and per-call observation length, with only a few tasks lying near either extreme.
> > >
> > > We conduct additional analysis suggests that the strongest overall predictor of difficulty is the total context pressure induced by these factors (Spearman r = -0.61, p = 0.026), which is consistent with the paper’s use of environment description length as the main difficulty metric. When we further decompose this pressure, per-call observation length shows a clearer association with performance degradation than interaction count alone: average observation length is negatively correlated with accuracy (r = -0.49, p < 0.001), whereas the number of tool calls by itself shows little standalone correlation (r = 0.08, p = 0.55).
> > >
> > > Therefore, our takeaway remains consistent with A1: in LOCA-bench, these two factors are usually coupled rather than cleanly separable, and the overall difficulty is best captured by environment description length. Within this overall context pressure, longer per-call observations appear to be the stronger contributor to degradation than interaction count alone.

---

### Official Review · Reviewer_2pDp · 2026-03-13

**Soundness:** 3
**Presentation:** 3
**Significance:** 3
**Originality:** 3
**Overall Recommendation:** 4
**Confidence:** 3

**Summary:**

- The paper proposes a novel benchmark to evaluate LLM models at long-context reasoning and decision-making, but where long-contexts are an artifact of interactions with an environment.
- To this end, the paper proposes "LOCA-Bench". A key novelty is *only* varying the environment description lengths while keeping all else fixed (e.g., task semantics). As a result, this enables ablating model performances purely resulting from the long-context aspect.
- On the evaluation side, the paper benchmarks both closed-weight (e.g., Claude-4.5, GPT-5.2) and open-weight models (e.g., DeepSeek-v3.2). They also analyze various failure modes e.g., weaker instruction following at longer contexts.
- Finally, the paper explores how to improve performance of the models by using different strategies (e.g., clearing thinking blocks, adding memory tool).

**Compliance With Llm Reviewing Policy:**

Affirmed.

**Key Questions For Authors:**

1. How can one determine that the 15 "seed" tasks have noise that is low enough to not introduce significant variations in the results?
2. Failure Analysis - is this from cherry-picked trajectories or do they generally hold?
3. How do we expect the trajectory lengths or tool-calls use vary for golden/expert trajectories?

**Limitations:**

I could not find a section addressing limitations.

**Strengths And Weaknesses:**

### Strengths
1. **Extensive evaluation**: The paper evaluates 7 different models (both closed- and open-weight) for their main results. Some configurations stress-test the context lengths (e.g., 100K-1M).
2. **Novel insights esp. for practitioners**: Some insights (e.g., Section 4.2) are particularly helpful for practitioners who want to understand the performance gains in long-context agentic regimes.
3. **Well-written**: Paper is generally well-written and easy to follow
4. **Static -> Dynamic Large-contexts**: As the authors rightly point out, long-context evals are generally performed in a one-shot NIH-like setting. While this explains away *step* level performance, agentic tasks require understanding decision making over growing contexts.

### Major Concerns

**1. Very few tasks**
- The benchmark itself comprises of 15 "seed" tasks. For each task, the authors generate multiple variations by changing environment description length and random seeds.
- As a result, much of the the analysis rests on the diversity/robustness/quality of the 15 tasks. An ill-specified/noisy task can introduce significant variations and alter the results.
- This raises some follow-up concerns: Was quality control performed on the the tasks? What is the variance of success rates over the tasks (e.g., there exists 1-2 tasks that always succeeds/fails)?

### Minor Concerns

**2. Failure Analysis - unclear methodology and frequency**
- I appreciate the authors highlighting multiple failure modes in Section 3.2. I believe this is a critical part of the paper and provides general take-aways in long-context regimes.
- However, the methodology to determine these failure modes are unclear. Each failure mode describes one particular failure example. Do these findings generally hold? For instance, it was observed over multiple tasks in the benchmark.

**3. Context Engineering - Overfit to product offerings**
- While the section on context engineering solutions is helpful to a practitioner, I believe it lacks contribution for an academic paper.
- My main concern is that it largely presents conclusions by running ablation studies of different proprietary features (e.g., memory tools) over proprietary models (e.g., Gemini). As a result, it primarily appears to benchmark commercial product offerings in the benchmark.

**4. Positioning results with golden trajectories - missing**
- The main results and detailed and I appreciate the writing, However, some hypothesis are unclear to position the results in Section 3.2.
- Specifically, since these are largely synthetically generated tasks with potential golden solutions/trajectories, how should one expect the curves in Fig. 3 to evolve?
- For instance, given only the environment description varies, should one expect the #tool calls to be constant across different description lengths?

---

> ### Author Rebuttal · Authors · 2026-03-30
>
> **Q1**: Was quality control performed on the the tasks? What is the variance of success rates over the tasks (e.g., there exists 1-2 tasks that always succeeds/fails)?
>
> **A1:**
> We manually checked each task during design to ensure it remained solvable at different environment description lengths. To further verify robustness, in the rebuttal we reran the 32K setting for DeepSeek-V3.2-Thinking and MiniMax-M2.1. For each model, we added two new runs and compared them with the original paper run. As shown below, the variance in success rate across runs is small.
>
> | Model                   | Run 1 | Run 2 | Run 3 | Mean  | Std (↓) |
> |-------------------------|------:|------:|------:|------:|--------:|
> | DeepSeek-V3.2-Thinking |  61.3 |  60.0 | 60.51 |  60.6 |    0.53 |
> | MiniMax-M2.1           |  42.7 |  41.3 |  42.7 | 42.23 |    0.66 |
>
> Moreover, up to 128K, every task is solved by at least one model. Only at 128K and beyond do some tasks become unsolved by all models such as filterlow_selling_products, update_material_inventory, and woocommerce_new_welcome. Manual inspection suggests these failures are still mainly caused by the increased complexity of the environment state. For example, on woocommerce_new_welcome, even GPT-5.2-medium wastes many tool calls on repeated WooCommerce pagination queries and BigQuery schema exploration, and fails to finish the task.
>
> **Q2**: However, the methodology to determine these failure modes are unclear. Each failure mode describes one particular failure example. Do these findings generally hold? For instance, it was observed over multiple tasks in the benchmark.
>
> **A2**: Thank you for the suggestion. During the rebuttal, we conducted a quantitative analysis of the failure modes. Please see our response to **Reviewer nxpc’s Q2** for details. The results show that these patterns are common across tasks.
>
> **Q3**: My main concern is that it largely presents conclusions by running ablation studies of different proprietary features (e.g., memory tools) over proprietary models (e.g., Gemini)?
>
> **A3**: Thank you for the suggestion. Many of the context-editing strategies we study, such as tool-result clearing, thinking-block clearing are also widely used in evaluation harnesses for open-source models [1]. Memory mechanisms are especially common for personalization, context management, and experience transfer in long-horizon agentic tasks [2,3]. In Section 4, we benchmark strong open-source models, including DeepSeek-V3.2-Thinking. During rebuttal, we additionally evaluate K2-thinking, with results shown below. This is consistent with our main conclusion that these strategies tend to help frontier models more than open-source models.
>
> | Model / Setting             | 128K |
> | --------------------------- | ---: |
> | K2-thinking                 | 0.08 |
> | + Programmatic Tool Calling | 0.13 |
> | + Context Awareness         | 0.11 |
> | + Tool-result Clearing      | 0.08 |
> | + Thinking-block Clearing   | 0.11 |
> | + Context Compaction        | 0.05 |
> | + Memory Tool               | 0.11 |
>
>
> [1] MemSkill: Learning and Evolving Memory Skills for Self-Evolving Agents
> [2] Memtool: Optimizing short-term memory management for dynamic tool calling in llm agent multi-turn conversations
>
> **Q4**: How should one expect the curves in Fig. 3 to evolve? For instance, given only the environment description varies, should one expect the #tool calls to be constant across different description lengths?
>
> **A4**: Thank you for the suggestion. In our setup, environment description length is the total length of tool outputs collected by scripts interacting with the environment, so increasing it directly increases the total amount of tool-output content. By contrast, trajectory length depends on the model’s own behavior, such as its reasoning and tool-use efficiency, and is therefore not an intrinsic property of the environment.
>
> We also estimated the number of tool calls using a procedure similar to that used for tool-output length. The results show that tool calls also increase with environment description length, especially at larger context sizes.
>
> | Environment description length |   8K |  16K |   32K |   64K |   96K |   128K |   256K |
> | ------------------------------ | ---: | ---: | ----: | ----: | ----: | -----: | -----: |
> | # Tool calls                   | 7.93 | 8.52 | 11.42 | 17.12 | 82.95 | 143.55 | 346.52 |
>
>
> Thus, increasing environment description length raises both tool-output volume and the number of tool calls. For example, in canvas-list-test at 128K, the environment contains 70 courses, and querying each course for quiz and assignment status requires roughly 110–130 tool calls per run. Meanwhile, several key responses are extremely long: list_courses returns about 69K characters, get_upcoming_assignments returns 118K–140K characters, and course_config.json is 89K characters. Together, the many interactions and large tool outputs create substantial context pressure.

---

> > ### Author Rebuttal · Reviewer_2pDp · 2026-04-02
> >
> > Thanks for the detailed clarification.
> >
> > Some follow-up questions:
> >
> > **(FQ1) variance over success rates ... run $n$**
> > Is each "run" is an independent run of the LLM on the *same* task? I'm asking because my primary concern was variance over different tasks keeping the initial task and environment description length fixed (what the paper in L209 reports as "five random seeds to produce distinct environments")
> >
> > **(FQ2) quantitative analysis over failures**
> > Thanks for reporting this. How were the "primary failure cause" determined? Given the trajectory length can be >100K tokens (Fig. 3) involving multiple turns and additionally errors can compound (or recovered mid-way and fail for a different reason later).

---

> > > ### Author Response · Authors · 2026-04-03
> > >
> > > **Q1:** variance over success rates ... run $n$ Is each "run" is an independent run of the LLM on the same task? I'm asking because my primary concern was variance over different tasks keeping the initial task and environment description length fixed (what the paper in L209 reports as "five random seeds to produce distinct environments")
> > >
> > >
> > > **A1:** We appreciate the clarification. Under each environment description length setting, our benchmark contains 15 tasks. For each task, we use 5  seeds to generate 5 distinct environment instances while keeping the task instruction and environment description length fixed, resulting in 15*5=75 evaluations per setting. The purpose of using multiple seeds is to reduce the variance of the reported success rate by averaging over multiple independently generated instances of the same task, rather than relying on a single seed, which may introduce substantial variance into the reported result.
> > >
> > > This should be distinguished from the two additional experiments we added in the rebuttal, where we reran the full 75-instance evaluation twice from scratch. Those extra runs measure rerun reproducibility, whereas the your question concerns seed-level variation under fixed task instructions and fixed environment description length settings.
> > >
> > > To address this point directly, we analyzed the success rate of each seed, aggregated over the 15 tasks for each model, as shown in the table. The main takeaway is that we do not observe one particular seed that is consistently and disproportionately bad enough to dominate the variance. Although the weakest seed is not identical for the two models, the broader pattern is the same: performance differences are distributed across seeds, rather than being caused by a single pathological seed. This is exactly why we average over 5 seeds—single-seed evaluation can deviate noticeably depending on the sampled instance set, while averaging across multiple seeds provides a more stable estimate of model performance.
> > >
> > > | Model | Seed 1 | Seed 2 | Seed 3 | Seed 4 | Seed 5 |
> > > |---|---|---|---|---|---|
> > > | DeepSeek-V3.2-Thinking | 66.7% (10/15) | 66.7% (10/15) | 66.7% (10/15) | 46.7% (7/15) | 60.0% (9/15) |
> > > | MiniMax-M2.1 | 40.0% (6/15) | 60.0% (9/15) | 33.3% (5/15) | 53.3% (8/15) | 20.0% (3/15) |
> > >
> > > **Q2:** quantitative analysis over failures Thanks for reporting this. How were the "primary failure cause" determined? Given the trajectory length can be >100K tokens (Fig. 3) involving multiple turns and additionally errors can compound (or recovered mid-way and fail for a different reason later).
> > >
> > > **A2:**  Thank you for the question. As you noted, a failed trajectory can involve multiple failure patterns, and any one of them may be enough to cause the task to fail. To make the analysis as objective as possible, we examine multiple runs of the same task, record the failure patterns that appear in each run, and count how frequently each pattern occurs across runs. We then identify the most frequently observed pattern as the primary failure mode for that task.

---

### Official Review · Reviewer_CcVw · 2026-03-13

**Soundness:** 4
**Presentation:** 3
**Significance:** 3
**Originality:** 4
**Overall Recommendation:** 5
**Confidence:** 4

**Summary:**

This paper introduces LOCA-bench, a novel benchmark for evaluating language agents under context growth. Unlike prior long-context benchmarks that mainly test static needle in a haystack tasks, LOCA-bench studies agentic settings where models must explore environments, use tools, and act under dynamically growing context. The benchmark scales context by enlarging environment states while keeping task semantics fixed, and is implemented with mock tool environments and execution-based evaluation. Across 15 seed tasks expanded to 525 samples, the authors show that agent performance drops sharply as context grows. The paper further analyzes characteristic long-context failure modes and evaluates several context-engineering strategies, finding that approaches such as programmatic tool calling can substantially improve performance.

**Compliance With Llm Reviewing Policy:**

Affirmed.

**Final Justification:**

The authors' follow-up addresses my core concern about decoupling context length from increased environment complexity. Since the underlying information space and exploration problem are held constant, the accuracy recovery can be attributed to context pressure specifically rather than to task difficulty. The benchmark addresses a timely problem and the context-engineering comparisons are practically useful. I would recommend Accept for this paper.

**Key Questions For Authors:**

1. How do you quantify and control the confounding between "increased environment complexity" and "pure context length increase"? Have you conducted controlled experiments where the number of distractors remains fixed while context length is increased through padding with redundant information, to separate these two factors? Such an ablation would significantly strengthen the core claims.
2. Have you conducted systematic fidelity validation of mock servers, e.g., comparing API responses with real services? Are there cases where mock server simplification led to agent behavior different from real scenarios?

**Limitations:**

yes

**Strengths And Weaknesses:**

Strengths:
1. The degradation of agent performance under dynamically growing context during real-world tool-call-intensive interactions is an under-explored critically important problem.
2. Through template-based environment generation and parameterized configuration, LOCA-bench can scale context length to arbitrary sizes while keeping task semantics fixed. Other agent benchmarks may produce long trajectories but do not study context length as a controlled variable.
3. The paper systematically compares the effectiveness of various context management strategies, providing practical guidance.

Weaknesses:
1. Tasks appear to primarily follow a "retrieve and organize information from Canvas/Email/databases" pattern. While service variety is broad, the core cognitive pattern may be relatively homogeneous.
2. While the paper claims to keep task semantics fixed, increasing environment description length (more courses, more emails) inherently enlarges the information space, potentially increasing both the number of distractors and exploration complexity. This means observed performance degradation may not be entirely attributable to context rot but partly to the task itself becoming more complex.

---

> ### Author Rebuttal · Authors · 2026-03-30
>
> **Q1:** Tasks appear to primarily follow a "retrieve and organize information from Canvas/Email/databases" pattern. While service variety is broad, the core cognitive pattern may be relatively homogeneous.
>
> **A1**: Figure 2 illustrates a task that retrieves and organizes information from Canvas, Email, and databases, but we would like to clarify that this example is only meant to demonstrate our task-generation pipeline, and our benchmark is substantially broader. As shown in Table 7, the full task set spans diverse scenarios and server combinations, reflecting a much wider range of patterns. Many tasks go beyond information retrieval, and some require quantitative analysis followed by conditional decision-making. For example, in ab-testing, the agent must compute per-scenario conversion rates from BigQuery data and use those results to identify the winner. Others require processing unstructured documents and generating new content, such as apply-phd-email and payable-invoice-checker. We also include tasks that require temporal reasoning across sources, such as cross-source anomaly detection in machine-operating.
>
> **Q2**: While the paper claims to keep task semantics fixed, increasing environment description length (more courses, more emails) inherently enlarges the information space, ...... performance degradation may not be entirely attributable to context rot but partly to the task itself becoming more complex.
>
> **A2**: During task construction, we increased the amount of added information in a balanced way, expanding both the useful content and the adversarial content that acts as noise. The core logic of the task remains the same: the model must still identify the correct information despite the interference from noisy information. Although “task complexity” can be defined in different ways, our design follows the same general idea as needle-based benchmarks such as RULER [1] and Michelangelo [2], which increase useful information while also introducing more irrelevant information to create interference.
>
>
> [1] Hsieh C P, Sun S, Kriman S, et al. RULER: What's the real context size of your long-context language models?[J]. arXiv preprint arXiv:2404.06654, 2024.
>
> [2] Vodrahalli K, Ontanon S, Tripuraneni N, et al. Michelangelo: Long context evaluations beyond haystacks via latent structure queries[J]. arXiv preprint arXiv:2409.12640, 2024.
>
> **Q3**: Have you conducted systematic fidelity validation of mock servers, e.g., comparing API responses with real services? Are there cases where mock server simplification led to agent behavior different from real scenarios?
>
> **A3**: We performed iterative fidelity validation for every mock server against its real-service counterpart, with the goal of minimizing mismatches when the real APIs could be successfully called. For each server, we ran parallel sessions in which a Claude Code agent invoked the same tool with the same parameters on both the real and mock servers, and then compared the returned response structure, status codes, error messages, and field values. Human reviewers also examined the agent trajectories to identify possible hallucinations or mismatches that automatic comparison might miss. Each server underwent 4–10 rounds of refinement, and we considered it aligned only after a full validation round showed no differences under successful real-API calls.
>
> At the same time, we note that real API services can be unstable and may occasionally fail for reasons such as network traffic or other unexpected conditions. Our mock servers do not attempt to reproduce this kind of service instability, since our goal is to evaluate agents in a stable and controlled environment rather than to model transient infrastructure failures. Below, we present several representative alignment cases.
>
> | Compared Aspect                   | Service     | Example                                          | Real vs. Mock Behavior                                                              |
> | --------------------------------- | ----------- | ------------------------------------------------ | ----------------------------------------------------------------------------------- |
> | Error messages                    | Canvas      | `get-course(course_id=99999)`                    | Both return `Error: Canvas API Error (404): The specified resource does not exist.` |
> | Response structure                | Canvas      | `create-assignment(course_id=1, name="Test")`    | Both return `200` with a JSON assignment object                                     |
> | Error messages / Status codes     | WooCommerce | `remove-product(productId=999999)`               | Both return `Error: WooCommerce API error: Invalid ID. (Status: 400)`               |
> | Response structure / Field values | WooCommerce | `remove-product(productId=<valid>, force=true)`  | Both return `200` with a deleted product object                                     |

---

> > ### Author Rebuttal · Reviewer_CcVw · 2026-04-04
> >
> > I really appreciate the clarification and detailed explanation. I think my concerns for Q1 and Q3 have been fully resolved.
> >
> > However, my concern about environment complexity and pure context length still remains partially addressed. While I agree the provided analogy at a high level, RULER and Michelangelo are mainly static retrieval benchmarks, whereas LOCA-bench specifically targets agentic settings with dynamic exploration. In the agentic setting, adding more entities (courses, emails, products) might enlarge the exploration space, requiring more tool calls and more complex navigation strategies. Ablation experiments might help strengthen the core claim that the performance degradation is due to context length rather than increased task complexity.

---

> > > ### Author Response · Authors · 2026-04-06
> > >
> > > We thank the reviewer for the thoughtful follow-up. We agree that, in an agentic setting, increasing the number of entities can enlarge both the exploration space and the amount of information accumulated in context, so these factors are not cleanly separable. LOCA-bench is designed to study precisely this coupled regime: the agent starts with limited knowledge, must explore the environment through tool use, and continually incorporate newly discovered information into its context. Thus, in our setting, exploration burden and context growth naturally co-evolve, while the underlying task objective remains fixed. （Line 77-80）
> > >
> > > That said, our results suggest that the degradation cannot be explained solely by the enlarged exploration space. In Table 2, we fix the environment at 128K, so the underlying information space and exploration problem remain unchanged, and only reduce effective context usage during execution. Under this controlled setting, strategies such as programmatic tool calling consistently improve accuracy, and context-editing methods such as tool-result clearing and context compaction recover performance in several cases by freeing context budget. This provides evidence that, although exploration difficulty and context growth are coupled by design in LOCA-bench, context pressure itself is a substantial contributor to the observed degradation.

---

### Decision · Program_Chairs · 2026-04-30

**Decision:**

Accept (regular)

**Comment:**

Reviewers agree that LOCA-bench addresses a timely and critical problem: evaluating language agents under dynamically growing contexts during real-world, tool-call-intensive interactions [CcVw, 2pDp].
By parameterizing environment states, the benchmark allows for controlled scaling of context length while keeping underlying task semantics fixed, which offers valuable insights into context rot in agentic settings.

While the initial evluation received generally positive scores, the review panel raised several valid concerns.
Reviewers CcVw and nxpc noted that adding more entities inherently enlarges the exploration space, which confounds pure context length with task difficulty.
Reviewer 2pDp questioned the robustness of the 15 seed tasks and requested a deeper failure mode analysis and also pointed out that context engineering strategies were primarily evaluated on proprietary models.

In the rebuttal, the authors provided controlled ablations demonstrating that context pressure remains a primary driver of performance degradation.
They also supplied quantitative failure breakdowns, rerun variance statistics showing low variance across random seeds, and additional open-source model evaluations (e.g., K2-thinking).
The reviewers found the responses convincing. Specifically, Reviewer CcVW has raised their score from 4 to 5 after rebuttal.
Given the high quality of the benchmark design, the practical insights for long-context agents, and the comprehensive author responses, the recommendation is to accept the submission.